# Pathology of Initial Axon Segments in Chronic Inflammatory Demyelinating Polyradiculoneuropathy and Related Disorders

**DOI:** 10.3390/ijms232113621

**Published:** 2022-11-07

**Authors:** Edyta Dziadkowiak, Marta Nowakowska-Kotas, Sławomir Budrewicz, Magdalena Koszewicz

**Affiliations:** Department of Neurology, Wroclaw Medical University, Borowska 213 Str., 50-556 Wroclaw, Poland

**Keywords:** juxtaparanodal region, initial axon segment, axonal transport vesicles, chronic inflammatory demyelinating polyradiculoneuropathy

## Abstract

The diagnosis of chronic inflammatory demyelinating polyradiculoneuropathy (CIDP) is based on a combination of clinical, electrodiagnostic and laboratory features. The different entities of the disease include chronic immune sensory polyradiculopathy (CISP) and autoimmune nodopathies. It is debatable whether CIDP occurring in the course of other conditions, i.e., monoclonal IgG or IgA gammopathy, should be treated as a separate disease entity from idiopathic CIDP. This study aims to evaluate the molecular differences of the nodes of Ranvier and the initial axon segment (AIS) and juxtaparanode region (JXP) as the potential cause of phenotypic variation of CIDP while also seeking new pathomechanisms since JXP is sequestered behind the paranode and autoantibodies may not access the site easily. The authors initially present the structure of the different parts of the neuron and its functional significance, then discuss the problem of whether damage to the juxtaparanodal region, Schwann cells and axons could cause CIDP or if these damages should be separated as separate disease entities. In particular, AIS’s importance for modulating neural excitability and carrying out transport along the axon is highlighted. The disclosure of specific pathomechanisms, including novel target antigens, in the heterogeneous CIDP syndrome is important for diagnosing and treating these patients.

## 1. Introduction

According to the European Academy of Neurology/Peripheral Nerve Society guideline on diagnosis and treatment of chronic inflammatory demyelinating polyradiculoneuropathy (CIDP) from 2021, autoimmune nodopathies are not classified as CIDP. Antibodies to nodal–paranodal cell adhesion molecules such as contactin-1 (CNTN1), contact-related protein 1 (Caspr1), neurofascin-155 (NF155) and neurofascinNF140/186 isoforms were detected in a small subgroup of patients meeting the 2010 EFNS/PNS criteria for CIDP. It has now been proposed to call these conditions “autoimmune nodules” and not to consider them as variants of CIDP, as they have distinct clinical features without overt macrophage-mediated demyelination or inflammation [1]. To date, autoimmune polyneuropathies, based on the presence of specific antibodies and electrophysiology, have been classified as primarily demyelinating or axonal. However, recent studies have challenged the concepts of demyelinating and axonal changes in nerve conduction and the basis of the classic dichotomous classification. Thus, acute neuropathies with anti-ganglioside antibodies, classified as axonal and resulting from nodal dysfunction, may present with reversible conduction failure and rapid recovery. It seems an apparent paradox that neuropathies labelled as axonal can be rapidly reversible. Additionally, autoimmune disorders targeting specific molecules of the nodal region, although not showing pathological evidence of demyelination, may show electrophysiological changes characteristic of demyelinating neuropathy. Uncini et al. propose to classify autoimmune neuropathies based on the myelin fiber domains involved (i.e., the site and molecular target of the autoimmune attack) and based on the antigen when known [2].

In patients with suspected CIDP, the main objective of electrodiagnostic testing is to demonstrate a demyelinating pattern, which can be very difficult to obtain in the case of a long-standing disease process when axonal involvement becomes evident. The second revision of the CIDP criteria (EAN/PNS 2021) now allows the diagnosis of chronic inflammatory polyradiculoneuropathy (CIP) to be confirmed using clinical features and response to immunoglobulin/plasma exchange treatment. Randomized controlled trials have confirmed the efficacy of immunosuppressive/immunomodulatory treatment in patients with CIP. In these patients, electrophysiological studies find axonal lesions that do not meet electrophysiological criteria for demyelinating lesions [1,2]. Some authors have recently proposed that autoimmune peripheral neuropathies be subdivided into several subtypes depending on the clinical phenotype, pathophysiology and neurophysiological features—“chronic internodopathy; (CIDP)” and “nodoparanodopathy” [3,4,5]. Oh et al. presented diagnostic criteria they proposed specific to another type of autoimmune neuropathy, called “Chronic Inflammatory Axonal Polyneuropathy” (CIAP). Nevertheless, this entity needs to be confirmed by independent studies [6].

The aim of the study was to evaluate, based on the current data, the pathomechanism of chronic inflammatory polyradiculoneuropathy, as well as the understanding of the anatomy of the axon, juxtaparanode structure, novel antigens, and in consequence the understanding of CIP pathophysiology and possible treatment targets. The available evidence base was analyzed, finding molecular differences in the nodes of Ranvier and the AIS and highlighting fundamental differences in the mechanisms that control ion channel distribution at the nodes of Ranvier and the AIS. The authors discuss the problem of whether the damage to the juxtaparanodal region, Schwann cells and axons is still a separate disease entity or still CIDP (Figure 1).

## 2. Methods

### 2.1. Search Strategy

The authors conducted a literature search focused on the various phenotypes of CIDP, with a particular interest in the characteristics of a paranodal type of the disease. The search engines PubMed via MEDLINE and Google Scholar were used from the beginning of 2010 until 31 July 2022. Reviews and research studies were included, with further revision of their reference lists for their relevance to the topic. Conference abstracts and papers written in languages other than English were excluded. The used keywords: chronic inflammatory demyelinating polyneuropathy, CIDP, CIAP, CIP, variants, anatomy, physiology, phenotypes, immunology, inflammatory process, nodal and paranodal antibodies, trafficking mechanism, axon initial segment, AIS, Schwann cell, protein 4.1B. In addition to using particular keywords, the authors used PubMed Advanced Search Builder to find the most relevant records. Two analysts (ED and MNK) worked separately to find the most pertinent papers by screening the search engines.

### 2.2. Data Extraction

As a result, 230 records were identified and screened separately by ED and MNK (Figure 2). Each analyst prepared a list of records that were relevant to the study. Then, these record lists were double-read by both, and 122 were found suitable and their full-text manuscripts were acquired. Both analysts independently read all manuscripts.

### 2.3. Qualitative Analysis and Synthesis

Researchers worked independently and prepared a list of relevant full-text manuscripts, followed by a comparison of both lists and discussion. The studies most relevant to the study and included in this review comprised 98 publications.

## 3. Anatomy

Several factors play a role in the excitability of neurons and conduction of stimuli. For the action potential initiation, the axon initial segment (AIS) plays a crucial role, with several factors modulating excitability, such as localization, anatomy and channels. Due to the postulated importance of dysfunction in this area for the pathophysiology of atypical forms of CIDP, the structure of the AIS is presented [7,8].

For the rapid conduction of action potentials, the formation of myelin around the axon is critical. The Schwann cells develop myelin with microvilli surrounding areas of axon in the peripheral nervous system. The myelin may be divided into the node of Ranvier (NOR), the paranodal junction (PNJ) and the juxtaparanodal region (JXP). They are covered by the basal lamina (BL) [7].

A detailed review of current findings on myelin sheath structure in NOR and PNJ has been presented in another study [8]; now, the authors intend to discuss JXP in detail, which could be essential for the development of atypical forms of peripheral nervous system pathologies.

### 3.1. Axon Initial Segment

The AIS is a dynamic structure, with changing geometry, usually located just after the neuron hillock (Figure 3). It has several microscopic morphological features: bundles of 3–10 microtubules called fascicles, a dense layer of finely granular material undercoating the plasma membrane, and dispersed ribosomes [8,9,10]. It is characterized by the lowest threshold for action potentials within neurons due to the high density of voltage-gated Na+ (Nav) channels (for review, Debanne et al., 2011 [11]). Those channels interact with a molecular complex composed of membrane scaffolds, cell adhesion molecules and cytoskeletal proteins [12]. 

The core of this structure is ankyrin G (ANK3), a member of the family of proteins that mediate the attachment of membrane proteins to the cytoskeleton, concentrating specifically at the AIS and nodes of Ranvier [15,16,17]. Other ankyrins, G and R, are found in the Ranvier junction, while B is localized mainly in the internode region of the axon [16,17]. Due to its modular structure, ankyrin G organizes the AIS scaffold [18]. The ANK3 with its aminoterminal side is inserted into the actin–spectrin submembrane scaffold, and it anchors AIS-specific membrane structures, such as Nav1, Kv7 and Kv1 channels as well as neurofascin 186 (NF-186) and NrCAM [13,17,19]. Conversely, the ANK3 connects with microtubules via microtubule-associated proteins, such as EB1/3 and Ndel1 [20]. With the accumulation of ANK3, the motion of molecules in the AIS membrane decreases, leading to the formation of a barrier between axonal and somatodendritic compartments [21]. Another important structure in the compartmentalization of the neuron is the submembrane cytoskeleton called the membrane-associated periodic skeleton (MPS) [22]. It consists of long filaments of actin organized in the braid pattern, forming ring-like structures connected with alpha- and beta-spectrin and other associated molecules [22,23]. MPS is present in the central and peripheral nervous system and plays a role in the mechanical resistance of axons [24,25]. membrane filtering at the AIS [21], maintenance of axonal microtubules [26] and regeneration of axons [23].

Spectrins in neurons consist of α2- and β2-spectrin tetramers, which assemble axially along the axons and parts of dendrites [22]. β4-spectrin, encoded by the SPTBN4 gene, is expressed in nodes and AIS [27]. This group of proteins forms MPS and enables organelle transport, promoting axon stability and allowing axon growth [28].

Other proteins, such as microtubule cross-linking factor 1 (MTCL1) and tripartite-motif containing 46 (TRIM46), are one of the earliest markers of axonal initial segment formation with a crucial role in the stabilization of microtubules and regulation of cargo trafficking [29,30]. Additionally, end-binding proteins (EB1 and EB3) anchor microtubules in the region of AIS, where microtubules, actin and associated proteins, including Tau and microtubule-associated protein 2 (MAP2), cooperate in maintaining an intracellular filter [28].

### 3.2. Juxtaparanode Region

The JXP is adjacent to the PNJ, and its organization depends upon the barrier formed by the PNJ on the one side and the unique JXP membrane complex and its linkage to the cytoskeleton on the other (Figure 4) [16,31].

The JXP contains the Shaker-like K+ channel α subunits Kv1.1 and Kv1.2 and the cytoplasmic Kvβ2 subunit [32,33,34]. K_v_1.1 channels are Shaker-related delayed-rectifier channels, which are present both in the PNS and central nervous system (CNS) [35]. This isoform was described in the juxtaparanodal regions, at branch points of myelinated motor axons and in basket cells of the cerebellum [34,35,36]. K_v_1.2 channels, also members of the Shaker K+ channel family, are voltage-gated potassium channels, which play a crucial role in maintaining neuronal excitability [37,38,39].

K channels associate with Caspr2 (contactin-associated protein-like 2), formatting Caspr2/K+ channel complexes [40]. Caspr2 is found in both the CNS and PNS, expressed in juxtaparanodes and some isolated paranodal loops. It has been postulated that its role in stabilizing potassium channels and thus maintaining the resting potential of the membrane may be of clinical importance in several disorders, such as autism or Parkinsonian-ataxia spectrum [39,40,41,42,43]. The extracellular region of Caspr2 is a mosaic of domains, including laminin G, EGF-like cysteine-rich domain, and fibrinogen domains. In addition, intracellularly it has a 4.1 binding site and a binding site for an abundant protein–protein interaction module called PDZ (type 2) [40,44,45,46,47].

On both the axon and glial cell sides, the presence of TAG-1, a cell adhesion molecule (also called Contactin2), was stated in juxtaparanodes [48,49]. The localization of Kv1 channels at the JXP strongly depends on both Caspr2 and TAG-1, suggesting that axon–glia interactions mediated by these proteins allow myelinating glial cells to organize ion channels in the underlying axonal membrane [50].

Other molecules playing crucial roles in the integrity of the juxtaparanode are proteins from the 4.1 family, from which sensory neurons express three proteins (4.1B, 4.1N and 4.1R) in the peripheral nervous system [51,52,53,54], and the fourth (4.1G) is expressed by Schwann cells [45]; they participate in clustering Kv1.2 channels, Caspr2 and Contactin2 in juxtaparanodes [53]. Intracellularly, protein 4.1B and α2/β2 spectrin bind to Caspr2. This transmembrane glycoprotein has laminin G-like domains, EGF-like cysteine-rich domains, a Pro-Gly-Tyr repeats (PGY) motif, an amino-terminal discoidin domain and a 4.1 binding domain [44,45].

Transmembrane receptors—ADAM22 and ADAM23—integrate with proteins from the LGI (Leucine-rich glioma-inactivated) family and participate in axon–glial cell communication and myelination during PNS development. They also recruit the PSD-93 (postsynaptic density protein 93) and PSD-95 (postsynaptic density protein 95) [54,55,56]. The latter are members of the membrane-associated guanylate kinase (MAGUK) family and bind to Kv1 channels without Caspr2 mediating this process [54].

### 3.3. Transport of Components

In myelinated axons, the junctions with Schwann cells at the paranodes prevent the lateral diffusion of membrane molecules and channels [57]. Thus, the replenishment of axonal membrane components requires sorting in axonal transport vesicles and delivery to the target axon sections, such as nodal, internodal or juxtaparanode regions, and clearance from the mistargeted domains [58]. One of the possible mechanisms used for the active transport of molecules such as Nav1 and Neurofascin186 along microtubules is ANK3, which can interact with kinases (Kif5, casein kinase CK2), proteins EB1/EB3 and spectrin. Upon reaching its destination, ANK3 dissociates from Kif5 to leave a cargo [56,57]. Anterograde transport of Kv1 channels, due to their heteromeric architecture, is more complicated and requires binding of the Kvβ2 subunit with the microtubule motors, such as Kif3/kinesin-2 and protein EB1 [59,60]. In this case, the signal of detachment in the AIS region is probably provided by cyclin-dependent-kinase phosphorylation [61].

Caspr2 and Tag-1/Contactin-2, and the other pair ADAM22 and ADAM23—juxtaparanodal components—can be sorted in the same transport vesicles. However, there is probably a different mechanism for the release of these proteins, as TAG-1 dominates the juxtaparanodal regions and Caspr2 is more evenly distributed along the axon [56,62].

Other mechanisms can actively remove molecules that were not transported to the proper compartment. Such a mechanism has been described for Caspr2, for which transport is not polarized and can also lead toward dendrites. In this case, PKC-phosphorylation of the 4.1B binding site results in endocytosis of Caspr2 and further transport towards the axon using the Kif3 motor [63,64].

## 4. Pathology of Initial Axon Segments

### 4.1. Pathology of Initial Axon Segments in CIDP

Based on studies in animal models, incomplete regeneration of damaged axons has been demonstrated in CIDP through the impaired pro-regenerative function of Schwann cells, which are affected by inflammatory mediators. In the peripheral nervous system, regeneration of damaged axons is mainly promoted by Schwann cells, which undergo transdifferentiation from myelinating cells to Büngner repair cells, which are repair Schwann cells that populate the distal site of axon damage and promote axonal regeneration and regeneration of the target tissue. In Schwann cells, a concentration of N-cadherin promotes the formation of multicellular cords (Büngner bands) responsible for guiding regenerating axons through the damaged area. Trophic factors and cytokines secreted by repair Schwann cells and macrophages promote axonal repair and regeneration. Schwann cells alter their morphology, down-regulate myelin genes and simultaneously up-regulate genes responsible for axon growth, neuronal survival and macrophage invasion. Schwann cells are controlled by regulatory factors, including negative regulators of myelination, such as c-Jun, p57kip2 and Sox-2, and positive regulators of myelination, such as Oct-6, Krox-20 and Sox-10. This phenomenon may explain the clinical observation of incomplete recovery in patients with CIDP despite optimal immunosuppressive treatment [65,66,67,68,69,70]. In contrast, Tzekova and coworkers found that IVIG positively influences the differentiation and maturation of Schwann cells and increases their potential to induce axonal growth [71]. 

In the case of axon damage, rapid changes in a large set of regeneration-associated genes expression occurs, enabling axonal formation, elongation and guidance [72]. These genes encode such important factors as c-JUN [72], STAT3 [73], SOX11 [74] or SMAD1 [75]. Degeneration manifests by the loss of MPS, and independently of this process, the activation of caspase apoptotic pathways [76]. The mechanisms by which the MPS reorganizes itself in the outgrowing or regenerating axon are not fully understood, although some evidence for the accumulation of spectrin in the growth cone and the accelerated formation of spectrin periodicity has already been gathered [23].

One of the main pro-inflammatory factors in CIDP is tumor necrosis factor α (TNFα). It is a pleiotropic cytokine that can induce various cellular effects, from proliferation and differentiation to apoptotic cell death. Recent experimental studies have partially elucidated the cellular pathways activated by TNFα through binding to its receptors, type I (TNFRI) and type II (TNFRII). The TNFRI receptor has the potential to induce apoptosis. At the same time, TNFRI and TNFRII are independently capable of activating the transcription factors NF-κB and c-jun, which can influence cell survival and differentiation. Nuclear factor κB is a dimeric complex of cytoplasmic proteins displaying DNA-binding activity and nuclear translocation properties. Activating stimuli disrupt its constitutive, inhibitory binding to IκB proteins, enabling the translocation of the active nuclear factor kappa-light-chain-enhancer of activated B cells (NF-κB) heterodimers into the nucleus, where its p65, c-Rel and RelB subunits regulate the transcription of target genes. The nuclear protein c-jun is a key member of the pathway that regulates the response of neurons and glial cells to cellular injuries. Induction of c-jun and its transcriptional activity includes a cascade of molecular kinases leading to phosphorylation in the nucleus of N-terminal c-jun kinases (JNKs), which in turn activate their target, c-jun [77,78,79,80]. Joshi and colleagues [65] found that Schwann cells exposed to CIDP sera demonstrated significantly reduced mRNA expression of the genes p57kip2 and c-Jun compared to Schwann cells exposed to control sera in vitro. Additionally, this study found that loss of Schwann cell support was associated with lower serum granulocyte-macrophage colony-stimulating factor (GM-CSF) levels in CIDP and correlated with altered expression of c-jun and p57kip2 in Schwann cells. Inactivation of these regulatory factors resulted in altered expression of neurotrophins, including glial cell lineage-derived nerve growth factor (GDNF), brain-derived neurotrophic factor (BDNF) and nerve growth factor (NGF) in CIDP-conditioned Schwann cells in vitro. Bonetti and co-authors found that increased signals for NF-κB molecules, but not c-jun, were observed via immunohistochemistry on nuclei of Schwann cells in nerve biopsies from CIDP patients compared to controls [79].

Single nucleotide polymorphisms (SNPs) in TAG-1, which is a crucial molecule for axon–Schwann cell interactions, are related to the IVIg responsiveness of CIDP patients [81].

Anticontactin-1 IgG3 autoantibodies are rare but detected during the acute onset of autoimmune neuropathies. There is evidence that anticontactin-1 prevents adhesion interaction and that chronic exposure to IgG4 anticontactin leads to structural changes in the nodes accompanied by neuropathic symptoms. However, the pathomechanism of the acute onset of the disorders and the pathogenic role of anticontactin-1 IgG3 is mainly unknown. Doppler et al. conducted an animal study where Lewis’ rats were intraoperatively injected with the IgG of patients with anticontactin-1 autoantibodies. Patient IgG obtained during the acute onset of the disorders (IgG3 dominance) and IgG from the chronic phase of the disorders (IgG4 dominance) were comparatively studied. Conduction blocks were measured in rats injected with “acute” IgG more often than after injection of “chronic” IgG (83.3% versus 35%). There was no demonstrated dispersion of paranodal proteins or sodium channels to juxtaparanodes, as observed in patients after chronic exposure to anticontactin-1. The authors conclude that autoantibodies to the paranodal protein contactin-1 IgG3 were associated with acute onset of disorders in the patients studied. This is becauseanti-contactin-1 IgG3 induces an acute conduction block that is most probably mediated by autoantibody binding and subsequent complement deposition. These findings support the concept of anti-contactin-1-associated neuropathy as a paranodopathy with the nodes of Ranvier as the site of pathogenesis [82].

As a guardian of action potential initiation, the AIS is ideally positioned to modulate the excitability and plasticity of the neuron. Recent studies have found that the anatomical properties of the AIS (e.g., length, position along the axon, etc.) can be dynamic and plastic in response to normal developmental or pathological activity [81,82,83,84].

Autoimmune disorders may disrupt the AIS. For example, mice with inflammatory demyelination show profound disruption and even loss of the AIS [85,86].

### 4.2. Pathology of Initial Axon Segments in Related Disorders

The main difference between the peripheral nervous system (PNS) and central nervous system (CNS) nodes is the interaction partners of neurofascin 186 (NF186). Neurofascin 186 is the cell adhesion molecule that anchors voltage-gated sodium channels in the node and AIS. It is a transmembrane protein with six immunoglobulin (Ig)-like domains, four fibronectin type III (Fn) domains, and one mucin domain. Its binding partner is ankyrin G inside the cell, which in turn interacts with βIV-spectrin. In the PNS, NF186 interacts extracellularly with the soluble gliomedin and NrCAM that are located on the Schwann cell microvilli that fill the nodal space whereas in the CNS, NF186 interacts with extracellular matrix proteins [44].

In demyelinating lesions of multiple sclerosis (MS), junctional disruption is associated with a heterogeneous distribution of Nav channels with diffuse immunoreactivity, sparse broad aggregation and a diffuse distribution of paranodal and juxtaparanodal elements. The early paranodal alterations occur as observed at the border of MS lesions in periplaques and normal-appearing white matter (NAWM), with the Kv1 channels abutting or even overlapping the nodal region. Loss of juxtaparanodal clusters is observed, with extensive Kv1.2 expression along denuded axons [14,85,87].

Genetic mutations in the AIS components (i.e., ankyrins, ion channels and spectrins) and injuries may cause different neurological disorders. All forms of ankyrin proteins (ankyrin-R, ankyrin-B and ankyrin-G) expressed in different tissues are vital for normal tissue functioning. Disorders or injuries could alter AIS structure: focal cortical and white matter stroke, hypoxia-induced ischemic injury, Angelman syndrome, Alzheimer’s disease, multiple sclerosis. Altered AIS structure and function are associated with schizophrenia, epilepsy, (spectrinopathy?) congenital myopathy, neuropathy and deafness, among others (for review, Lorincz and Nusser, 2008 [83,84]).

Mutations in the SPTBN4 gene, which encodes β4-spectrin, are connected to myopathy, neuropathy and auditory deficits in humans [88,89]. Knierim et al. presented a patient with congenital myopathy, deafness and neuropathy who had a homozygous nonsense mutation in SPTBN4 [c.1597C>T, NM_020971.2; p.(Q533*), NP_066022.2; ClinVar SUB2292235] encoding β4-spectrin, a non-erythrocytic member of the β-spectrin family. The authors cautioned that a larger number of patients need to be described to confirm the triad of congenital myopathy, neuropathy and deafness as the defining symptom complex for β4-spectrin deficiency [90].

Previous studies have found a strong correlation between AIS defects and Alzheimer’s disorder [91]. Sun et al. found increased expression of miR-342-5p, resulting in decreased expression of AnkG, and impairing the localization of Nav1.6 channels in the AIS [92].

Many of the ion channel subunits associated with mutations in epilepsy are localized to the AIS, making this site a hotspot for epileptogenesis. Dynamic regulation of Kv and Nav channel expression and localization in the AIS results in altered excitability, suggesting that activity-dependent modulation of trafficking to these sites may affect neuronal and network function. Among the Kv1 and Kv2 α and Kvβ subunits known to be associated with AIS, only the KCNA1 gene encoding the Kv1.1 α subunit has mutant alleles associated with neurological disorders, in particular episodic ataxia type 1 (EA-1), often associated with seizures, which result from loss of function of Kv1.1-.containing channels [82,91,92,93].

Nascimento et al., based on a series of experiments using a chronic constriction injury (CCI) model of neuropathic pain that produced abnormal spontaneous activity in dorsal root ganglion (DRG) neurons, demonstrated that AISs generate the pathological spontaneous activity underlying neuropathic pain [90].

The TRIM46 autoantibody-associated paraneoplastic neurological syndrome has also been described [90]. Van Coevorden-Hameete et al. [94] described three patients with autoantibodies to several epitopes of the AIS protein TRIM46. These patients had various manifestations of neurological deficits—progressive encephalomyelitis, cerebellar ataxia, rapidly progressive dementia. Two of these patients were diagnosed with small-cell lung carcinoma (SCLC). Other case reports of paraneoplastic syndromes associated with β4-spectrin autoantibodies have also been reported in the literature, e.g., in a patient with a single breast cancer who developed motor neuropathy [94,95].

Heterozygous nonsense mutations in the α2-spectrin gene, SPTAN1, have been identified in patients with juvenile-onset hereditary motor neuropathy and in patients with epilepsy [96,97,98].

## 5. Conclusions

The disclosure of specific pathomechanisms, including novel target antigens, in the heterogeneous CIP syndrome is of major importance for the diagnosis and appropriate treatment of these patients. Discovery in the detail of cell adhesion molecules that form a complex with other adhesion molecules, ion channels and the cytoskeleton to form the nodal and paranodal compartments changed understanding and treatment of chronic inflammatory polyradiculoneuropathy (CIP). Currently, chronic immune sensory polyradiculopathy (CISP) and autoimmune nodopathies are not classified as CIDP. Likewise, the discoveries of the AIS and juxtaparanodal region have the potential to have an effect on the field of inflammatory neuropathies. A detailed understanding of the structure of the axon, juxtaparanode area and transport of components may, in the future, make it possible to modify the classification of autoimmune polyneuropathies and clarify the term CIDP to only refer to disorders involving the node of Ranvier and Schwann cells.

## 6. Future Directions

The authors would like to indicate that the presented data are not without gaps, which should be filled in research in the near future. The major challenges in future research relate to:The anatomy of the axon initial segment: the node of Ranvier, the paranodal junction and the juxtaparanodal region, together with understanding the principles of axonal transport of various factors important in the physiology and pathology of peripheral nerves.The disclosure of specific pathomechanisms important in the peripheral nerve damage, including novel target antigens and their mutual dependence and dependence on the presence of other pro and anti-inflammatory factors.The reference of autoimmune disturbances to clinical findings, which may lead to the creation of a new classification of inflammatory neuropathies.All of the above-mentioned scientific challenges should result in new therapeutic options tailored to the individual patient.

## Figures and Tables

**Figure 1 ijms-23-13621-f001:**
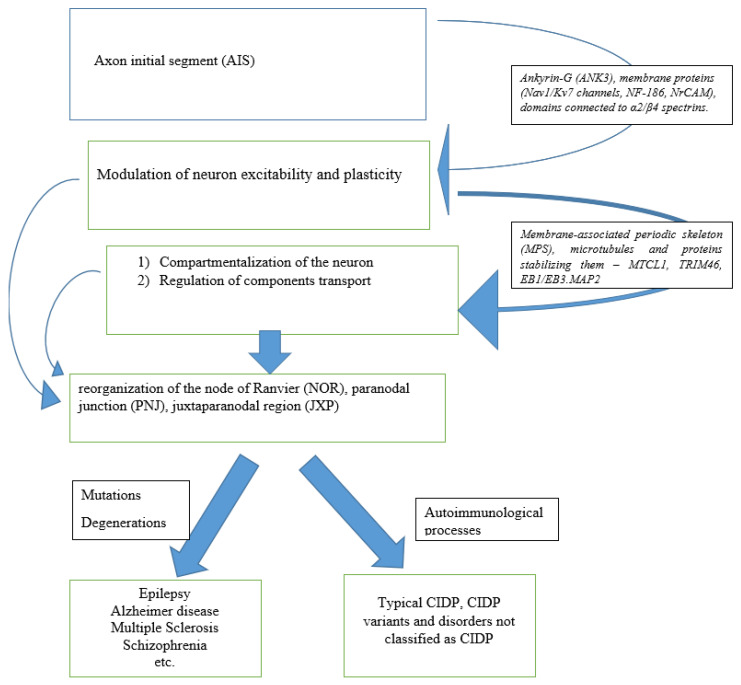
Graphic representation of the pathological mechanisms of the initial axonal segments in disorders of the central nervous system and peripheral nervous system.

**Figure 2 ijms-23-13621-f002:**
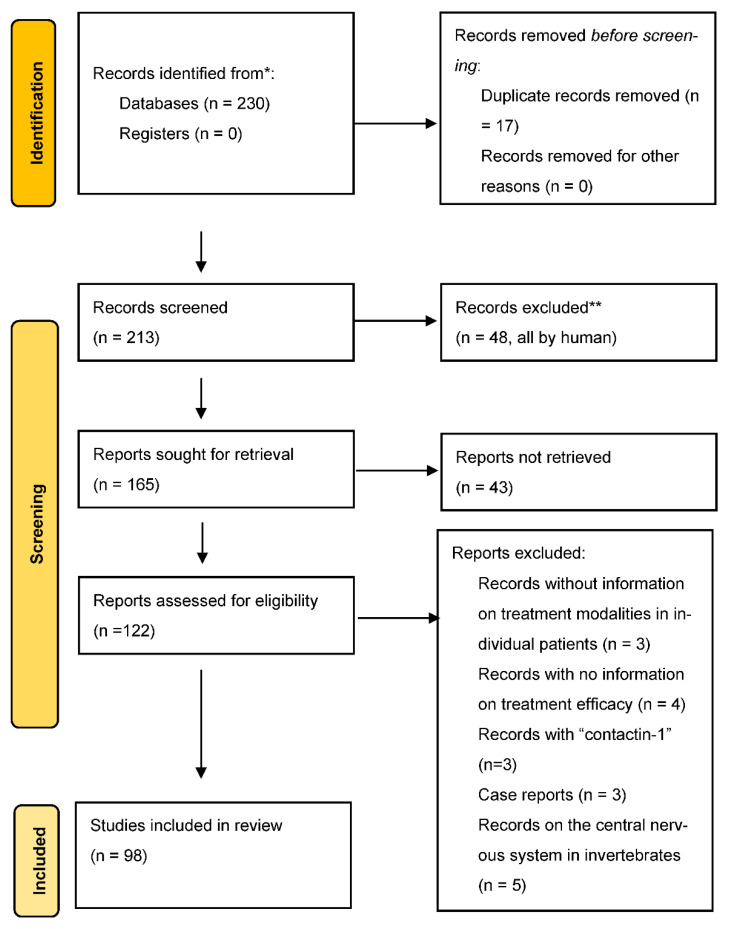
Schedule of the selection of publications. * Consider, if feasible to do so, reporting the number of records identified from each database or register searched (rather than the total number across all databases/registers). ** If automation tools were used, indicate how many records were excluded by a human and how many were excluded by automation tools.

**Figure 3 ijms-23-13621-f003:**
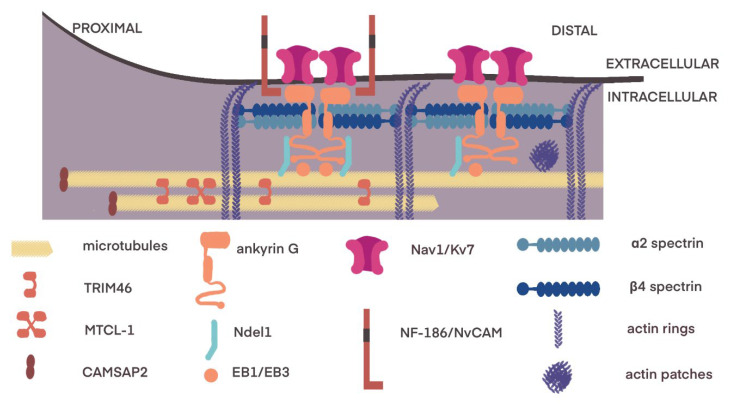
Axon initial segment Ankyrin-G (ANK3) anchors several membrane proteins (Nav1/Kv7 channels, NF-186, NrCAM) and with other domains is connected to α2/β4 spectrins. Actin rings consisting of long filaments organized in braid pattern are connected with spectrin. Actin may also assemble in actin patches. ANK3 binds to microtubules via EB1/EB3 proteins and Ndel1. TRIM46 and MTCL-1 cross-link microtubules. Modified according to [13,14].

**Figure 4 ijms-23-13621-f004:**
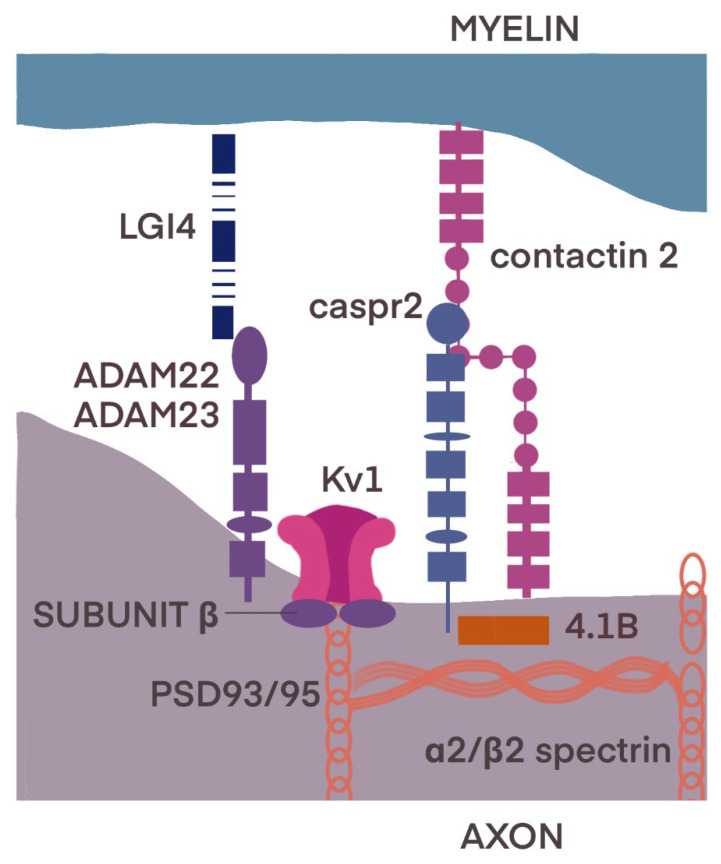
Molecular organization of juxtaparanodal region. The voltage-gated Kv1.1/1.2 channels consisting of α and cytoplasmic β subunits are associated with Caspr2 forming Caspr2/Kv complexes. Contactin2/TAG-1 localized on axon interacts with Contactin2/TAG-1 on the myelin. Transmembrane receptors ADAM22 and ADAM23 integrate with proteins from the LGI4 and recruit PSD-93/95. Intracellularly, protein 4.1B and α2/β2 spectrin bind to Caspr2. At the internode, the axonal cell adhesion molecules Necl1 and Necl2 interact with 4.1B and also interact with glial Necl4. Modified according to [16,31].

## Data Availability

No new data were created or analyzed in this study. Data sharing is not applicable to this article.

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
