# Peer review of "Pathology of Initial Axon Segments in Chronic Inflammatory Demyelinating Polyradiculoneuropathy and Related Disorders"

_ijms, 2022, doi:10.3390/ijms232113621_

Round 1

Reviewer 1 Report

This review manuscript discusses the status of our contemporary understanding of the pathogenic mechanisms inherent in chronic inflammatory demyelinating polyradiculoneuropathy (CIDP). The authors focus on key pathological hallmarks of CIDP and discuss whether idiopathic CIDP is a distinct disease when compared to CIDP triggered by know etiogenic factors (MGUS, diabetes, cancer). In their discourse presented in this review manuscript, the authors raise pertinent questions about the clinical criteria for differential diagnosis of CIDP from other closely related spectrum of autoimmune polyneuropathies. The following minor suggestions may increase the merit of the manuscript. 

1. The review, in its entirety, could be structured better. Specifically, Section 4 discussing the underlying pathogenic mechanisms, could be restructured such that it confers a relatively more lucid and clear understanding of the underlying cellular and molecular mechanisms that are distinctly implicated in CIDP, and not other closely related spectrum of autoimmune polyneuropathies. 

2. The Conclusions could be more specific. The Conclusion section could be expanded and the authors could expatiate and discuss the specific cellular and molecular hallmarks as well as specific pathogenic mechanisms that are distinctly involved in CIDP versus other closely related spectrum of autoimmune polyneuropathies. 

3. The authors could add a separate "Future Directions" section to highlight and emphasize the significance and novelty of the findings presented in this review. 

4. The Abstract could be rephrased and restructured such that it better embodies the content of the review. 

Author Response

Thank you for your response and the relevant comments regarding some aspects of our study. Please find enclosed explanations to your remarks:

  1. Figure 1 has been added as a graphic representation of the pathological mechanisms of the initial axonal segments in disorders of the central and peripheral nervous system.
  2. Modified the Conclusions.
  3. The authors would like to point out that the data presented are not without gaps, which should be filled in research in the near future. A Section 6 "Future Directions" has been added, in which the main challenges in future research are highlighted.
  4. The Abstract has been restructured.

Appropriate changes were made and highlighted in the revised version of the manuscript. Hopefully you will find them satisfactory.

Reviewer 2 Report

In the paper by Edyta Dziadkowiak et al., titledPathology of initial axon segments in chronic inflammatory demyelinating polyradiculoneuropathy and related disordersthe authors reviewed the problem whether the damage to the juxtaparanodal region, Schwann cells and axons is still a separate disease entity or still CIDP.

The article is written on an intriguing and up-to-date topic of modern neuroscience and has a scientific interest.

However, several issues need to be addressed:

Major issues:

1.       In the introduction section, the authors are kindly invited to state the purpose of your research more clearly (a graphical representation of the main ideas), in order to improve the perception of the article.

2.      The reviewer recommends supplementing the article with a general final outline about the role of the damage of the juxtaparanodal region, Schwann cells and axons in CIDP.

3.      Please confirm whether permission to use the figures has been obtained from the authors (Figure 2, 3). Original figures are welcome.

4.      It is recommended to supplement the conclusions and make them more structured.

Minor issues:

Line 127 – reference

Line 135 – 3.1 missing the title of the paragraph

Little things that need to be fixed.

Can be accepted after careful revision.

Author Response

Thank you for your response and the relevant comments regarding some aspects of our study. Please find enclosed explanations to your remarks:

  1. Figure 1 has been added as a graphic representation of the pathological mechanisms of the initial axonal segments in disorders of the central and peripheral nervous system.
  2. The authors would like to point out that the data presented are not without gaps, which should be filled in research in the near future. A Section 6 has been added, in which the main challenges in future research are highlighted.
  3. The authors confirm that they have obtained permission to use the figures from the author (Figures 3 and 4).
  4. Modified the conclusions.
  5. Minor issues corrected.

Appropriate changes were made and highlighted in the revised version of the manuscript. Hopefully you will find them satisfactory.

Round 2

Reviewer 2 Report

The paper can be accepted in the current form.